# Role of MicroRNAs in Signaling Pathways Associated with the Pathogenesis of Idiopathic Pulmonary Fibrosis: A Focus on Epithelial-Mesenchymal Transition

**DOI:** 10.3390/ijms23126613

**Published:** 2022-06-14

**Authors:** Ana Ruth Cadena-Suárez, Hilda Arely Hernández-Hernández, Noé Alvarado-Vásquez, Claudia Rangel-Escareño, Bettina Sommer, María Cristina Negrete-García

**Affiliations:** 1Laboratorio de Biología Molecular, Instituto Nacional de Enfermedades Respiratorias (INER) “Ismael Cosío Villegas”, Calz. Tlalpan 4502, Col. Sección XVI, Mexico City 14080, Mexico; anah.cdn@gmail.com (A.R.C.-S.); hilda.0237@gmail.com (H.A.H.-H.); 2Departamento de Bioquímica, Instituto Nacional de Enfermedades Respiratorias (INER) “Ismael Cosío Villegas”, Calz. Tlalpan 4502, Col. Sección XVI, Mexico City 14080, Mexico; nnooee@gmail.com; 3Departamento de Genomica Computacional, Instituto Nacional de Medicina Genómica, Periférico Sur 4809, Col. Arenal Tepepan, Mexico City 14610, Mexico; crangel@inmegen.gob.mx; 4Escuela de Ingenieria y Ciencias, Tecnológico de Monterrey, Epigmenio González 500, San Pablo 76130, Mexico; 5Departamento de Investigación en Hiperreactividad Bronquial, Instituto Nacional de Enfermedades Respiratorias (INER) “Ismael Cosío Villegas”, Calz. Tlalpan 4502, Col. Sección XVI, Mexico City 14080, Mexico; bsommerc@hotmail.com

**Keywords:** miRNAs, idiopathic pulmonary fibrosis, EMT, myofibroblasts

## Abstract

Idiopathic pulmonary fibrosis (IPF) is a chronic and progressive disease with high mortality and unclear etiology. Previous evidence supports that the origin of this disease is associated with epigenetic alterations, age, and environmental factors. IPF initiates with chronic epithelial lung injuries, followed by basal membrane destruction, which promotes the activation of myofibroblasts and excessive synthesis of extracellular matrix (ECM) proteins, as well as epithelial-mesenchymal transition (EMT). Due to miRNAs’ role as regulators of apoptosis, proliferation, differentiation, and cell-cell interaction processes, some studies have involved miRNAs in the biogenesis and progression of IPF. In this context, the analysis and discussion of the probable association of miRNAs with the signaling pathways involved in the development of IPF would improve our knowledge of the associated molecular mechanisms, thereby facilitating its evaluation as a therapeutic target for this severe lung disease. In this work, the most recent publications evaluating the role of miRNAs as regulators or activators of signal pathways associated with the pathogenesis of IPF were analyzed. The search in Pubmed was made using the following terms: “miRNAs and idiopathic pulmonary fibrosis (IPF)”; “miRNAs and IPF and signaling pathways (SP)”; and “miRNAs and IPF and SP and IPF pathogenesis”. Additionally, we focus mainly on those works where the signaling pathways involved with EMT, fibroblast differentiation, and synthesis of ECM components were assessed. Finally, the importance and significance of miRNAs as potential therapeutic or diagnostic tools for the treatment of IPF are discussed.

## 1. Idiopathic Pulmonary Fibrosis (IPF)

Idiopathic pulmonary fibrosis (IPF) is a progressive, chronic, and devastating interstitial lung disease of unclear etiology with few therapeutic options and an average survival of 3–5 years after diagnosis [1]. Numerous factors have been implicated in the onset and development of this disease. Among them, reactive oxygen species (ROS) are very relevant; since they are associated with the low activity of the antioxidant systems (e.g., superoxide dismutase, catalase, or glutathione system), they support the role of oxidative stress in the physiopathology of IPF [2,3]. An enzymatic system associated specially with oxidative stress in IPF is the NADPH oxidase family [4]. This oxidase family is stimulated by transforming growth factor beta (TGFβ1), which increases superoxide synthesis and favors oxidative stress, promoting lung fibrosis [5]. It is worth mentioning that Fierro-Fernández et al. [6] studied miRNAs involved in redox regulation in lung fibroblasts of patients with IPF, and found that miR-9-5p had an inhibitory effect on TGFβRII and NADPH4 expression. Moreover, in a bleomycin (BLM) BLM-fibrosis mice model, the over-expression of miR-9-5p reduced the fibrogenesis, and its inhibition abrogated its anti-fibrotic effect in both in vivo and in vitro assays.

The mitochondria plays an important role in the generation of ROS and in the apoptosis process; it has also been implicated in IPF disease [7]. Recently, Bueno et al. [8] reported that alveolar epithelial cells type II (AECIIs) from lungs of IPF patients exhibited an accumulation of dysfunctional mitochondria, a fact associated with upregulation of endoplasmic reticulum (ER) stress. In this work, the authors also reported that PTEN-induced putative kinase 1 (PINK1) deficiency results in swollen and dysfunctional mitochondria, as well as defective mitophagy, which promotes fibrosis in the aging lungs. Additionally, an increase in ROS levels generated by mitochondria induces lung fibrosis by positive feedback between ROS and TGFβ activities [9].

ER stress is another proposed mechanism for the onset and progression of fibrotic diseases, including IPF [3]. Evidence shows that the expression of surfactant protein A2 mutants (SP-A2, SFTPA2) associated with pulmonary fibrosis induces the production of proteins that cannot be secreted and therefore accumulate in the cytoplasm, leading to protein instability and ER stress [10]. In addition, ER stress and unfolded protein response (UPR) have been associated with IPF through alveolar epithelial cells (AECs) apoptosis, epithelial-mesenchymal transition (EMT), and M2 macrophage polarization [11,12].

Hypoxia is another prominent clinical feature of IPF disease. However, its real role in this pathology is still poorly understood. Mechanistically, hypoxia stimulates the proliferation of fibroblasts through hypoxia-inducible transcription factor alpha (HIF-1α), and HIF-2α [13]. It has been suggested that, in fibroblasts, HIF-1α targets pyruvate dehydrogenase kinase and switches the glucose metabolism of the cells to glycolysis, inducing the myofibroblast differentiation [14]. Interestingly, in 2014 Bodempudi et al. [15] demonstrated that miR-210 stimulates the proliferation of IPF fibroblasts in response to hypoxia.

On the other hand, the contribution of endothelial mesenchymal transition (EndMt) process to some pathologies like IPF has been recognized lately. The first evidence showing that lung capillary endothelial cells could originate fibroblast through EndMT in a BLM-fibrosis mice model was performed by Hashimoto et al. in 2010 [16]. It was recently described that an endothelial cell (EC) dysregulation together with aberrant epithelial activation or autoimmune illnesses may evolve from Interstitial Lung Disease (ILD) to progressive pulmonary fibrosis [17]. Additional recent findings have revealed that, while the number of ECs decreases in the lung microenvironment, the number of fibroblasts and myofibroblast increases during the development of IPF, suggesting a key role of EndMT in this process [18]. It has been suggested that the activation of the TGFβ signaling pathway is common to almost all ailments associated with EndMT, including IPF. Moreover, the ECs that undergo EndMT possess disrupted tight cell-cell junctions while mesenchyme-specific factors like N-cadherin, alpha smooth muscle actin (α-SMA), fibronectin and vimentin, among others, are upregulated, contributing to tissue fibrosis progression [3,19]. In this regard, an interesting study in systemic sclerosis demonstrated for the first time that oxidative stress associated with the activity of NADPH-oxidase drives fibrosis and EndMT [20].

All the mechanisms mentioned above have been associated with IPF’s pathogenesis, and for a more complete evaluation of these, we recommend reviewing the excellent work of Giang Phan et al. [3]. Nevertheless, the aim of the present review is to analyze the function of microRNAs in the signaling pathways associated with pathogenesis mechanisms involving mainly epithelial mesenchymal transition (EMT).

In addition to the mentioned, the incidence and prevalence of IPF increase with age, and the disease is more common in male patients above 65 years of age [21]. Initially, IPF was considered as a chronic inflammatory disorder characterized by lung fibrogenesis [22]. However, the lack of response to potent anti-inflammatory therapy and the possibility to develop inflammation-independent fibrosis suggested that inflammation is not an important pathogenic event in IPF [23]. New evidence indicates that this disease represents epithelium-driven fibrosis, where the bronchioalveolar epithelium secretes a wide variety of growth factors, cytokines, chemokines, and matrix metalloproteinases. All these events induce the migration, proliferation, and activation of mesenchymal cells, a fact associated with the progressive destruction of the lung parenchyma [24,25]. IPF is also associated with environmental and genetic risk factors, aging-related processes, and profibrotic epigenetic reprogramming [26,27]. In addition, in 2008 an increase in the expression of genes associated with lung development was reported, suggesting that an aberrant activation of embryonic signaling pathways involved in epithelial-mesenchymal communication and epithelial cell plasticity participates in IPF [28]. Today, it is well known that epigenetic mechanisms such as microRNAs play a critical role in the genesis and progression of this disease [29,30,31]. Therefore, a better knowledge of these signaling pathways and their regulation by miRNAs would improve our comprehension of the mechanisms involved in IPF pathology and might help us propose more effective therapeutic strategies for its treatment.

## 2. MicroRNAs (miRNAs)

miRNAs are a group of single-stranded non-coding RNA with a length of 19–25 base pairs. The miRNAs are transcribed in the cell nucleus by RNA polymerase II as primary miRNAs (pri-miRNAs) [32]. Then, under the regulation of RNase III and Drosha, they are processed to form 70–80 nucleotides precursor miRNAs (pre-miRNAs) which are transported from the nucleus into the cytoplasm, via exportin-5 complex and the nuclear protein Ran-GTP. In the cytoplasm, this complex is processed by Dicer into a mature miRNA duplex [33]. A mature miRNA strand forms, and then the miRNA-induced silencing complex (miRISC) combines with the Argonaute protein and pairs with the “seed region” to target mainly the 3’UTR of different mRNAs [34], controlling their expression through its degradation or repressing its translation [35,36]. A large number of these miRNAs conserve target sites that play important roles during cell development and whose absence may lead to death [37,38]. In addition, it is well known that miRNAs are cell, organ, and tissue-specific [39,40], and it has been estimated that more than 60% of human coding genes are regulated by miRNAs [38]. At present, there is overwhelming evidence that miRNAs intervene in the regulation of cellular processes such as apoptosis, proliferation, and cell differentiation [34,41,42,43]. On the other hand, it is known that the aberrant expression of some miRNA can lead to the development of diseases like fibrosis and cancer [44,45,46].

## 3. MicroRNAs and IPF

Interestingly, in 2010 differences in the miRNAs expressed in lung tissues of IPF patients were demonstrated for the first-time, in comparison with miRNAs found in the lung of healthy subjects [47,48]. A year later it was reported that these deregulated miRNAs were associated with some developmental pathways that involved TGFβ1, Wnt, sonic hedgehog, p53, as well as VEGF and their complex regulatory networks [49]. Today, many studies are focused on evaluating the expression levels of profibrotic and antifibrotic miRNAs, both in the onset and progression of IPF; however, the results obtained still show contradictory expression levels of these miRNAs [50,51,52,53]. It is probable that these variations are related to the complexity of this lung disease, since it involves multiple genes and signaling pathways [29,54,55]. In addition, the intricacy of the regulatory process where several miRNAs might regulate a single gene, or numerous genes can be affected by a single miRNA, hinders the better comprehension of the mechanisms involved [56,57]. Currently, IPF is a disease with a poor prognosis because it has a low life expectancy, and it is difficult to diagnose. Therefore, studies evaluating the function of miRNAs involved in the regulation of signaling pathways during the progression of IPF could help us elucidate the mechanisms involved in its development and pathogenesis, consequently enabling the discovery of new early diagnostic biomarkers or novel approaches for IPF therapy.

In this review, we carried on searches on PubMed by using the following cues: “miRNAs and idiopathic pulmonary fibrosis (IPF)”; “miRNAs and IPF and signaling pathways (SP)”; and “miRNAs and IPF and SP and IPF pathogenesis”. Through this search method, we mainly analyzed articles published between 2016 and 2021. However, because of their relevance, articles published in other years were mentioned too, although we focused principally on the articles obtained through the method described above. Moreover, we analyzed mainly all articles evidencing the impact miRNAs have on cell-signaling pathways involved with the IPF pathology.

The manuscript was organized to consider both the expression direction of each miRNA (up or down-expressed) as well as its pro-fibrotic or anti-fibrotic properties. In addition, the main signaling pathways were summarized, highlighting those linked to increased extracellular matrix (ECM) synthesis and to cell regulatory mechanisms. Moreover, the potential utility of each miRNA as a target for IPF treatment was also discussed.

## 4. Importance of Epithelial Mesenchymal Transition (EMT) in the Pathology of IPF

EMT regulates different processes during the early stages of development of most organisms. However, it also plays a role in adult organisms and in the pathology of certain diseases such as organ fibrosis and cancer [58,59]. EMT is a process mainly controlled by three families of transcription factors, namely, Snail1/2, Zeb1/Zeb2 and helix-loop-helix, which intervene in repressing E-cadherin expression and inducing mesenchymal gene expression [25,60,61]. Regarding IPF, it is characterized by presenting fibroblast foci localized in sub-epithelial layers close to areas of alveolar epithelial cell injury and repair, which promote abnormal epithelial-mesenchymal interactions and increase proliferation of fibroblast associated with an excessive synthesis of collagen and ECM [62]. In spite of all this information, the origin and activation process of fibroblasts/myofibroblasts during the pathogenesis of IPF remain undefined and controversial to this day. In 2005, Willis et al. [63], using TGFβ, showed for the first time the transition of alveolar epithelial cells (AECs) to myofibroblast through EMT. A year later, the mechanism involved with fibroblast accumulation during fibrogenesis in IPF was reported. In this last study, it was revealed that AECs were fibroblasts´ progenitors in vivo through EMT, and that ECM was a key regulator of this process [64]. Additionally, it is well known now that EMT is involved in the exacerbated fibroblastic response during the epithelial regeneration, a hallmark in the pathology of IPF [65]. The importance of AECs also derived from their role during the epithelial-mesenchymal switch, which induces alterations in their morphology, cellular architecture, and adhesion capacity, conferring on them a higher capacity for migration and resistance to apoptosis.

In addition to the above, an increase in both expression and synthesis of matrix metalloproteinases has been reported, with a consequent augment in extracellular matrix degradation that favors cell invasion processes [66].

## 5. Signaling Pathways Involved in the EMT Process

At present, numerous evidences indicate that aberrant activation of different developmental pathways are associated with the pathology of IPF [28,67,68,69]. EMT is a highly dynamic process that can be initiated by extracellular transcription factors or by the activation of different signaling pathways [59,60]. Some important inducers of EMT include fibroblast growth factor-2 (FGF-2) [70], epidermal growth factor (EGF) [71], insulin-like growth factor-II (IGF-II) [72], connective tissue growth factor (CTFG) [73], WNT ligands [74], and TGFβ [75]. However, TGFβ is considered an inducer of EMT in normal epithelial cells, as well as the principal profibrotic cytokine in IPF [63,66,75,76,77].

### 5.1. Smads Intracellular Effectors

Smad proteins intervene in the cell signaling acting as transcription factors. Eight Smad proteins have been reported in vertebrates (Smad1 to Smad8), and these are divided into three groups: Receptor-Smads (R-Smads), Common Smad (Co-Smad) and Inhibitory Smads (I-Smads) [78,79]. R-Smads are regulated through a receptor. Smad2 and Smad3, which are some examples, are activated through phosphorylation by TGFβ and Activin receptors, whilst Smad1, Smad5 and Smad 8 are activated by ALK receptor tyrosine kinase in response to bone morphogenic protein (BMP) or other ligands [78,79,80]. The R-Smad phosphorylation induces the formation of oligomeric complexes with Smad4, which in turn functions as common mediator Smad (Co-Smad) of R-Smads and is required to regulate the transcription of target genes [81,82]. R-Smads and Smad4 contain a conserved MH1 and C-terminal MH2 domain. In comparison, inhibitory Smads (Smad6, Smad7) lack the recognizable MH1 domain, but keep the MH2 domain. Smad6 and Smad7 inhibit TGFβ family signaling through binding of their MH2 domains to the kinase activity site of the type I receptor (e.g., ALK), thus preventing recruitment and phosphorylation of effector Smad complexes [79].

### 5.2. Smad-Dependent Signaling Pathways in EMT Induced by TGFβ1

Although three TGFβ isoforms have been identified in mammalians (β1, β2 and β3), it is known that TFGβ1 is the most closely related to the development of IPF [83,84]. It is important to highlight that both activation of Smad-dependent and non-Smad dependent TGFβ pathways have been implicated in the differentiation and survival of myofibroblast in the pathogenesis of IPF [85,86]. TGFβ is secreted in a latent form bound to the latency-associated peptide (LAP), and when released, TGFβ dimers form a complex with the type II TGFβ receptor (TGFβRII), which phosphorylates type I receptors of TGFβ (TGFβRI) [82]. This last complex can start the signal transduction as a Smad-dependent or non-Smad-dependent pathway [87]. In the Smad-dependent pathway, activated TGFβRI phosphorylates cytoplasmic Smad 2/3 transcription factors and translocates them into the nucleus [81,88]. Currently, we know that Smad4 facilitates this process and together with ZEB1/2 complex modulates the pro-fibrotic genes expression associated with the EMT and also regulates the differentiation, proliferation, and migration of fibroblasts, and the accumulation of collagen [86,89]. Moreover, the Smad 2/3 complex could inhibit the expression of E-cadherin, through transcription factors Snail1, Snail2, Notch, or Twist, resulting in the induction of the target genes α-SMA, collagen, plasminogen activator inhibitor-1 (PAI-1), and at the same time, the expression of mesenchymal proteins as N-cadherin, fibronectin, and metalloproteinases [77,90]. In contrast, others elements of the Smad family, such as Smad6 and Smad7, act as negative regulators by inhibiting the phosphorylation of Smad 2/3 complex with Smad4 or TGFβRI [79,91]. Moreover, inhibitory Smads can recruit E3 ubiquitin–protein ligases Smurf1 and Smurf2, which target Smad proteins for its proteasomal degradation, thereby blocking Smad 2/3 activation, facilitating receptor degradation, and terminating TGFβ/Smad signaling [92]. However, in spite of all this knowledge, the complexity of networks where Smad proteins are implicated is enormous, since these complexes activate or repress at the same time hundreds of target genes in the same cell, and under tightly controlled conditions (Figure 1A).

## 6. Non-Smad-Dependent Signaling Pathways in EMT Induced by TGFβ

The distinction between Smad-dependent and non-Smad-dependent pathways can be difficult to explain because there may be cross-talk between these pathways with non-Smad proteins [86]. Non-Smad-dependent pathways associated with TGFβ dependent EMT, include the activation of Erk, JNK, and p38 MAPK kinases pathways [79,93]. Moreover, signaling pathways that include PI3/Akt, Notch, RhoA, as well as growth factors and Wnt are involved too [86,94]. These pathways promote cellular changes during the EMT process, as tight/adherens junction disassembly, cytoskeletal rearrangement, repression of epithelial markers, and β-catenin nuclear translocation [63,94]. On the other hand, the non-Smad-mediated signaling pathways can interact with Smad-mediated genomic signaling through modulation and activation of different miRNAs [79,95]. For instance, the role of miR-200 in the induction of EMT was suggested for the first-time a decade ago, and involves its effect on transcription factors mRNAs such as ZEB1 and ZEB2 [96,97]. In addition, the role of miR-21as a regulator of Smad7 expression in the lung fibrosis was reported, too [48].

### 6.1. Erk, JNK, and P38 MAPK Pathways and Their Effects on the Development of EMT

The mitogen-activated protein kinase (MAPK) family includes three subfamilies: the extracellular signal-regulated kinases (Erk1 and Erk2), p38 MAP kinases, and c-Jun N-terminal kinases (JNKs) [92]. All these pathways contribute to EMT and may be activated by TGFβ, which regulates the expression of Smad proteins and intervenes in the Smad-independent TGFβ response [92,93,98]. For example, the activation of Erk 1 and Erk2 MAP kinases in response to TGFβ is initiated by Ras, which activates Raf and MEK 1/2 kinases. Moreover, EMT induced by TGFβ simultaneously activates Ras-ErK MAP kinase signaling [66]. The signaling by JNK and p38 MAPK is activated in response to several MAPK kinases kinases (MAPKKKs). Activation of MAPK pathways may affect the interaction between transcription factors as c-Jun or activating transcription factor 2 (ATF-2) with Smad protein, allowing the cross-talk between TGFβ-induced Smad and MAPK pathways [79]. Likewise, it has been reported that TNF receptor-associated factor 6 (TRAF6) is necessary for the Smad-independent activation of JNK and p38, as well as for its interaction with TGFβ receptors that activate TAK1, a kinase of p38 and JNK [99]. In contrast, the inhibition of ERK or p38 MAPK kinase activity represses EMT induced by TGFβ [93]. Moreover, JNK and p38 together with Smad pathways regulate the differentiation from fibroblasts to myofibroblasts induced by TGFβ [92] (Figure 1B).

### 6.2. PI3K/Akt/mTOR Pathway

The phosphatidylinositol-3-kinase (PI3K) pathway is a non-Smad pathway required for EMT during TGFβ-induced fibrosis. P13K induces mainly the profibrotic pathway of Akt/mammalian target of rapamycin (mTOR) [92]. mTOR is a typical serine-threonine- kinase that belongs to the PI3K related kinase family and interacts with several proteins to form two complexes: mammalian target of rapamycin complex 1 (mTORC1) and mTORC2 [100]. The heterodimer TSC1/TSC2 is a key upstream regulator of mTORC1 and functions as a strong stimulator in its kinase activity [101]. In epithelial cells that carry out EMT, TGFβ activates the P13/AKT pathway, causing the activation of these complexes [102]. The deregulated activation of mTORC1 is considered to be involved in the pathogenesis of fibrotic disorders by promoting the overproduction of collagen [92], while mTORC2 is required for cell migration and invasion in response to growth factors and the regulation of cell survival [101,103] (Figure 1C).

### 6.3. PI3K/Akt Signaling Pathway

The PI3K-Akt pathway is involved in cell growth, proliferation, survival and antiapoptotic processes. For its activation, PI3K stimulates the synthesis of phosphatidylinositol-3,4,5-triphosphate (PIP3), which recruits Akt and PIP3-dependent kinase that phosphorylates and activates Akt [68,104]. A mechanism suggested to induce fibroblast proliferation in IPF involves PI3/Akt. In this case, Akt phosphorylates and inactivates the expression of Forkhead Box O3 (FoxO3), a target gene of Akt in PI3K/Akt signaling pathway, and whose main function is to inhibit the cell cycle and to promote apoptosis [68,105]. On the other hand, PTEN, which has a critical role in regulating fibroblast removal during tissue repair, is considered the major negative regulator of the PI3K/Akt signal pathway, by inhibiting the Akt phosphorylation and promoting fibroblast apoptosis during tissue repair [106,107]. It has been described also that the regulation of expression genes involved with cell survival is regulated by p65, a subunit of NF-kβ, when Akt promotes its activation. On the other hand, caspase-9 can be phosphorylated by Akt inhibiting its activity and hindering cell death [104]. It is important to point out that the PTEN/PI3K/Akt signaling pathway is, moreover, a critical regulator of the connection between the TGFβ/Smad and Wnt/β-catenin pathways [28] (Figure 1C).

### 6.4. Rho-Like GTPases Regulated by TGFβ

This group consists of regulatory proteins of cytoskeleton organization and cell migration such as Rho, Rac and Cdc42. TGFβ regulates RhoA activity at the tight junctions of epithelial cells through the interaction with the TGFβRI/Par6 complex, allowing the phosphorylation of partitioning-defective protein (Par6) by TGFβRII in a specific serine. This process enhances RhoA ubiquitination by Smurf1 and degradation at tight junctions [66,108]. On the other hand, the activation of Rho kinase (ROCK) is the result of the RhoA action during the assembly of actin stress fibers in the TGFβ-induced EMT [109] (Figure 1D).

### 6.5. Wnt Signaling Pathway Associated with EMT

Using gene expression microarrays, the upregulation of several genes of the Wnt signaling pathways in the lungs of patients with IPF, in comparison with control lungs, was demonstrated for the first-time [25]. By then, only one study had reported the aberrant nuclear localization of β-catenin both in AECs and fibroblasts from the fibroblastic foci of lungs from IPF patients [110]. TGFβ1 synergizes with Wnt/β-catenin signaling pathway to induce EMT during the development of IPF [28,111]. Specially, the Wingless/integrase-1 (WNT) family controls a variety of developmental processes, which include cell fate specification, proliferation, polarity, and cell migration [112]. Classically, Wnt signaling is classified in two branches: (a) Canonical Wnt/β-catenin signaling and (b) Non-canonical β-catenin independent Wnt signaling [74]. The key role of the canonical Wnt pathway is the accumulation and translocation of the adherens junction associated-protein β-catenin into the nucleus [113] (Figure 2A). Without stimulation of Wnt proteins, cytoplasmic β-catenin is degraded by a “β-catenin destruction complex”, formed by glycogen synthase kinase-3, adenomatous polyposis coli protein, Axin, and α-Casein Kinase I (GSK-3/APC/Axin/CKI-α) [114]. Phosphorylation of β-catenin by CK1-α and GSK-3 mark it for its ubiquitination and subsequent proteolytic destruction by proteasomal degradation [74,113] (Figure 2B). In comparison, the Wnt activation develops by binding Wnt proteins to the domain of the Frizzled receptor family (FZD_1_-FZD_10_). In addition, some co-receptors are also required, such as the low-density-lipoprotein-related protein5/6 (LRP5/6) [112,114]. When cytoplasmic β-catenin is stabilized and translocated to the nucleus, it forms complexes with members of the T-cell factor/lymphoid enhancer factor (TCF/LEF), which regulate the transcription of Wnt target genes [113,115,116]. This pathway is involved in the regulation of cell differentiation and proliferation [74]. The non-canonical β-catenin independent Wnt signaling can be divided into the Planar Cell polarity pathway (Wnt/PCP) and Wnt calcium (Wnt/Ca_2_) pathway [115,116]. Both initiate with the binding of Wnt proteins (Wnt5a, Wnt4 and Wnt11 mainly) to the FZDs receptor on the cell membrane [117]. And, while polarity and cell movement are controlled through small GTPases of the Rho family, the Ca^2+^ pathway is regulated by heterotrimeric G proteins which control Ca^2+^ signaling [118]. Various studies have shown that dysregulation of β-catenin dependent or independent WNT signaling play a role in the development and progression of IPF [67,110]. In the epithelium, Wnt5a facilitates AEC2 to AEC1 differentiation, while in normal lung fibroblasts it promotes proliferation, increases fibronectin synthesis, and inhibits apoptosis induced by hydrogen peroxide [119]. In comparison, during wound healing and fibrosis, the WNT signaling pathway induces anti-apoptotic and pro-fibrotic phenotypes, promoting epithelial and myofibroblast differentiation, as well as fibroblast proliferation and survival, favoring collagen synthesis [120].

However, and in spite of all advances in our comprehension of idiopathic pulmonary fibrosis, current therapeutic options still have great limitations [121,122,123]. Therefore, the search, analysis, and discussion of new treatment options are necessary. Among these new options are the miRNAs because of their role as regulators of signaling pathways and cell functionality, which suggests that they probably play a role in this pathology [124,125]. The most recent information on dysregulated miRNAs implicated in the pathophysiology of IPF is depicted in the Table 1. The information is grouped according to the pro-fibrotic or anti-fibrotic role of each miRNA involved, its target of action, its expression level in the disease, and finally and most importantly, the signaling pathway affected.

## 7. MicroRNAs with Pro-Fibrotic Properties and with Regulatory Activity of Signaling Pathways Linked with IPF

Data obtained from the bleomycin (BLM) mice model or from fibroblasts stimulated with TGFβ showed pro-fibrotic activity of some miRNAs. An example is miR-21, which was the first miRNA studied in the BLM-induced fibrosis mice model in 2010 [48]. In this study, the authors reported for the first-time that miR-21 induces an increase of fibrogenic activity in pulmonary fibroblasts, through the expression of Smad7. However, another novel molecular upstream mechanism evaluated in relation to the function of miR-21 was recently reported, both in a BLM-fibrosis mice model, as well as in human embryonic lung fibroblasts (IMR-90) stimulated with TGFβ. In this study, the results showed that miR-21promotes lung fibrosis, as well as TGFβ1-induced ECM over-synthesis, after binding of TGFβ1 to TGFβ1R1, which activates the SMAD2/3 complex and the nuclear entry of SMAD4, facts that regulate both miR-21 expression and ECM protein expression [126]. On the other hand, in a pulmonary fibrosis rat model induced by intra-tracheal injection of bleomycin, it was observed that prodigiosin attenuates pulmonary fibrosis by inhibiting miR-410 and leading to the downregulation of the TGFβ1/ADAMTS-1 signaling pathway [127]. Another study identified miR-124 as a regulator of the differentiation from lung resident mesenchymal stem cells (LR-MSCs) to myofibroblasts induced by TGFβ1. Additional evidence showed that changes induced by TGFβ1 such as alterations in cell viability, proliferation, and consequent decreases in cell apoptosis were reversed by the blockage of this miRNA. Moreover, AXIN1 was identified as a new target for miR-124 to activate the Wnt signaling pathway [121]. Another miRNA evaluated in an IPF mouse model induced by bleomycin was miR-9; in this study, Dai et al. [128] reported that the over-expression of miR-9 negatively regulates ANO1, which enhances the activation of the TGFβ-Smad3 signaling pathway, aggravating inflammation, promoting proliferation, and inhibiting apoptosis of lung fibroblasts. Previously, miR-301a had been reported as an activator of two major inflammatory pathways in cancer (NF-kB and Stat3) [158]. Recently, Wang et al. [129] showed that miR-301a was over-expressed in the BLM-fibrosis mice model, in lung tissue of patients with IPF, and in normal and fibrotic fibroblasts stimulated with TGFβ. The authors demonstrated that the genetic deletion of miR-301a reduced the severity of lung fibrosis after bleomycin injection in the fibrotic mice model through a decrement of vimentin, α-SMA, and fibronectin expression. Furthermore, when they blocked miR-301a with an antagomir-301a, they observed a reduction in the proliferation and activation of lung fibroblasts as well as in the structural destruction of lung tissue in its experimental model. They showed that the negative regulation of its target TSC1 by miR-301a promoted the severity of lung fibrosis through the mTOR signaling pathway. Another recent study identified the profibrotic role of miR-424 in human lung fibroblasts (HLFs) stimulated with the TGFβ1, as well as in fibroblasts from IPF patients. The authors determined that this miRNA was involved in the increment of both α-SMA and CTGF protein expression, as well as in the myofibroblast differentiation. Additionally, it was also established, that the upregulation of miR-424 expression by TGFβ was SMAD3 dependent, acting as a positive regulator of the TGFβ signaling pathway by reducing the expression of Slit2 that exerts anti-fibrotic effects [130]. Given that LR-MSCs play an important role in pulmonary fibrosis, in 2020, Wang et al. [131] analyzed the miRNA and mRNA levels of LR-MSCs with or without TGFβ1 treatment. The microarray analysis made in the cells treated with this cytokine showed an overexpression of miR-152-3p, miR-140-3p, miR-148-3p, and miR-7a-5p all of which have as a target the Kruppel-like factor 4 (Klf4). Additionally, it was established that the inhibitor of growth family member 5 (ING5) was the common target for miR-34a-5p, miR-27b-3p, miR323-3p, miR-27a-3p, miR-34c-5p, miR-128-3p, and miR-224-5p, which were also overexpressed. The integrated miRNA/mRNA analysis showed that KLF4 and ING5 could be important targets for IPF treatment, due to their role as regulators in the myofibroblast differentiation and in the EMT process [131].

### MicroRNAs with Pro-Fibrotic Properties but Negative Regulators of Smad6/Smad7 Expression

Currently, it is well known that Smad7 blocks the function of Smad2/3, generating a negative effect on TGFβ1/Smad signaling [79]. For example, Wang et al. [132] determined in LR-MSCs treated with TGFβ an overexpression of miR-877-3p with Smad7 as its predictive target. When they developed functional studies with the mice bleomycin-treated model, miR-877-3p sequestration inhibited the differentiation of LR-MSC to myofibroblasts and attenuated pulmonary fibrosis by its effect on Smad7. Additionally, results obtained from human embryonic lung fibroblasts (HELF) stimulated with TGFβ, and lung tissue of a mice fibrosis model showed that miR-182-5p had profibrotic activities, as a consequence of the lower expression of its target gene Smad7 [133]. An analysis made with different databases showed that two miRNAs were significantly deregulated in IPF samples: miR-31, which was overexpressed, and miR-184, which was downexpressed. In order to study the molecular mechanisms involved, A549 epithelial cells were stimulated with TGFβ. Results showed a profibrotic role of miR-31 through downregulation of Smad6 and its failure to impede the phosphorylation of Smad2. In contrast, the antifibrotic role of miR-184 was confirmed when TGFβ abolished its function. In conclusion, the authors demonstrated a protection against TGFβ-induced fibrogenesis by repression of TGFβ-Smad2 and the TGFβ-P13-Akt signaling pathways, respectively [134].

## 8. MicroRNAs with Anti-Fibrotic Properties and with Regulatory Activity of Signaling Pathways Linked with IPF

Most of the recently reported miRNAs are within this group, since almost all are associated with the signaling pathways linked to TGFβ or with the activation of some of their components. A characteristic of this group of miRNAs is that they are important inducers of EMT, and are therefore considered the main contributors to lung fibrosis pathogenesis. In the middle of 2016, Ge et al. [135] showed some evidence pointing out that miR-323a-3p was downexpressed in the epithelium of IPF lungs and in the bleomycin-induced fibrosis model. In this same study, it was demonstrated that antagomirs for miR-323a-3p promoted murine lung fibrosis, and in contrast, when a miR-323a-3p mimic was used, the fibrosis was inhibited. Thus, the authors concluded that this miRNA attenuated the signaling associated with TGFα and TGFβ, as well as with the caspase-3 expression, by directly targeting the TGFα and Smad2 genes. The miR-29 has been considered an anti-fibrotic miRNA [49,159]. Therefore, the therapeutic effect of a single stranded RNA miR-29b match type (miR-29 PSh) was evaluated and compared with the effects of a double-stranded miR-29 mimic. In both fibrosis mice model and in cell culture experiments, miR-29b Psh-match showed lesser collagen synthesis than miR-29 mimic, a finding that supports its potential role as an effective therapeutic drug for pulmonary fibrosis [136]. The downregulation of miR-185 had already been associated with rapid IPF progression [160]. However, little is known about the expression of this miRNA in IPF bronchioalveolar lavage (BAL) cells and in alveolar macrophages (THP-1cells). The sub-expression of both miR-185 and miR-29a in this type of samples correlated with the disease severity and with eosinophil infiltration [137]. Furthermore, it was also observed that miR-185 down-regulation was associated with collagen deposition due to AKT pathway activation; while miR-29a downregulation caused the overexpression of the COL1A1 gene. The results obtained highlight the importance of evaluating the expression of these miRNAs in IPF BAL cells [137]. It had already been reported previously that Col5 was overexpressed in the lung tissue of IPF patients [161]; Col5 is regulated by miR-185 and miR-186 by a probable TGFβ signaling pathway activation [138]. On the other hand, the antifibrotic role of miR-130b-3p during the crosstalk between epithelial lung cells and fibroblast, a fact linked with insulin-like growth factor (IGF-1) expression in lung epithelium, was recently demonstrated [139]. Previous studies had already reported that the transcription factor from high mobility group AT-hook 2 (HMGA2) is induced by the TGF-β1/Smad3 signaling pathway during the EMT [47]. Additional evidence demonstrated also an essential role of miR-221 in EMT during liver fibrosis [162]. In this way, the downexpression of miR-221 and overexpression of HMGA2 in human IPF tissues and in bronchial epithelial cells treated with TGFβ was detected for the first time. Additionally, the authors observed EMT suppression when they transfected a miR-221 mimic in a mouse bleomycin-pulmonary fibrosis model, a fact derived from directly targeting HMAG2. This last fact supports the role of TGFβ1/Smad3 signaling pathway and its possible importance as a therapeutic target for the treatment of lung fibrosis [140]. Another miRNA that may have therapeutic applications in IPF is miR-1343, which is capable of reducing the expression of both TGFβRI and TGFβRII by directly targeting their 3′-UTRs in A549 epithelial cells and in normal lung fibroblasts stimulated with pro-fibrotic cytokine TGFβ [141]. Additionally, after TGFβ exposure, a significant increase in the levels of miR-1343 was observed and this phenomenon has been associated with an important decrease in pSmad2 and pSmad3 levels in the epithelial cells and fibroblasts, and with a reduction of fibrotic makers and a repression of EMT [141]. Another miRNA of relevance is miR-27a-3p whose expression is down regulated in lung fibroblasts from IPF patients in comparison to fibroblast from control subjects’ lungs. When this miRNA was over-expressed, pulmonary fibrosis induced in bleomycin-mice model was mitigated by targeting α-SMA, and Smad2/4 transcription factors. These results support the role of miR-27a-3p as a negative regulator of lung fibrosis since it inhibits the myofibroblast differentiation [142]. The expression of miR-27b was decreased in lung tissue of bleomycin-treated mice when compared to control mice, and in turn, the treatment of lung fibroblasts with bleomycin reduced the expression of this miRNA. Interestingly, its overexpression using a lentiviral vector in the LL29 human pulmonary fibroblasts stimulated with TGFβ1 inhibited the mRNA expression of collagen 3a1 and αSMA proteins, identifying TGF-βRI and Smad2 as their direct target genes. This result pointed out that miR-27b acts as an anti-fibrotic miRNA in pulmonary fibrotic fibroblasts [143]. On the other hand, Zhang et al. [144], reported that miR-18a-5p inhibits sub-pleural pulmonary fibrosis by targeting TGFβRII, and in consequence inhibits the TGFβ-Smad2/3 signaling pathway. Additional results reported by Huang et al. [145] showed that when a lentivirus expressing miR-18a-5p is injected into bleomycin-treated mice, pulmonary fibrosis and sub-pleural fibrosis were attenuated. An analysis made to evaluate the expression of miR-101 in lung tissues of IPF patients, from the Lung Tissue Research Consortium (LTRC), showed that this was one of the most downregulated miRNAs. In this way, the regulation of this miRNA and its cellular signaling was investigated. When miR-101was overexpressed in a fibrotic cell line (HLL29), the TGFβ-induced protein expression of α-SMA, COL1A1, and COL3A1 was inhibited. In contrast, with the transfer of anti-miR-101 to a normal lung, fibroblasts’ cell line increased the protein expression of these collagens and α-SMA. The results suggested that miR-101 inhibits the differentiation of fibroblasts to myofibroblasts stimulated by TGFβ via SMAD2/3 signaling pathway. Additionally, the effect of miR-101 overexpression through gene-transfer on the bleomycin mice model was examined. The results revealed a better lung function, with a reduction of COL1A1 and COL3A1, and of NFATc2, FZD6, and TGFβRI mRNA levels. Thus, the conclusion was that miR-101 is an anti-fibrotic miRNA with potential therapeutic properties to treat IPF [145]. In 2009, it was reported that miR-155 was induced by proinflammatory stimuli such as IL-1 and TNF-α in macrophages and dendritic cells [163,164]. Afterward, Pottier et al. [165] demonstrated, by in vitro functional assays, keratinocyte growth factor (KGF) as a new target for miR-155 in lung fibroblasts. Moreover, results obtained from a bleomycin-treated mice model showed that miR-155 expression levels in the lung fibroblasts correlated with the degree of lung fibrosis and depicted it as a potential key player during tissue injury. In contrast, in a recent study it was reported that miR-155 was downexpressed in human pulmonary fibroblasts stimulated with TGFβ. And when this miRNA was overexpressed to investigate its regulatory role in this cellular model, they observed attenuation of fibroblast proliferation, migration, and collagen synthesis. Furthermore, the authors observed a decrease in Smad1gene expression after miR-155 inhibition in this cellular model, by which they suggested an indirect miR-155-SMAD interaction linked to the TGFβ signaling pathway [146]. It had already been confirmed previously that miR-200a, miR-200b, and miR-200c were downregulated in mice with lung fibrosis induced by bleomycin [166]. However, during the early stages of acute respiratory distress syndrome (ARDS) induced in a mouse model by lipopolysaccharides (LPS), it was also shown that miR200b/c was downregulated, which was associated with an increment of their protein targets ZEB1 and ZEB2. Likewise, when a lentiviral packaged miR200b/c cDNA or ZEB1/sh RNA vectors were intratracheally administered, the pulmonary inflammation and fibrosis were reduced. Moreover, an increment of E-cadherin protein levels and a suppression of vimentin and αSMA protein expression were observed. These effects were associated with the inhibition of p38 MAPK and TGFβ/Smad3 signaling pathways [147]. To elucidate the role of some miRNAs, the miss-expressed miRNAs were firstly investigated by microarray expression, and afterwards their functions in activated lung fibroblasts were investigated. The authors found a miR-19a-19b-20a sub-cluster with the ability to suppress the activation in vitro of fibroblasts stimulated with TGFβ. Moreover, when this miRNA sub-cluster was intratracheally transferred into bleomycin-treated lungs, their results revealed not only the downexpression of pro-fibrotic genes such as ACTA2, COL1A1, or CTGF and Serpin, but also the over-expression of anti-fibrotic genes such as Dcn, Igfbp5, and MMP3 [148]. Another interesting example is miR-133a, which, in spite of being overexpressed in human lung fibroblasts (HLF) stimulated with TGFβ1, exhibited antifibrotic properties. To investigate whether this miRNA induced by TGFB modulated the fibroblasts differentiation into myofibroblasts, a miR-133a mimic was transfected into HLF. Surprisingly, the fibroblast differentiation was attenuated by a reduction of α-SMA expression. In contrast, miR-133a inhibitor improved TGFβ-induced myofibroblast differentiation. Additionally, target analysis and luciferase reporter assays showed TGFβRI, CTGF, and COL1A1 as direct targets of miR-133a. Moreover, in a functional analysis in a BLM fibrosis mice model, it was demonstrated that the overexpression of miR-133a ameliorated the lung fibrosis by a negative feedback regulator of TGFβ profibrogenic pathways [149]. In an interesting study, where the expression profile of miRNAs in exosomes obtained from BAL fluid (BALf) of elderly patients with IPF was analyzed, the downregulation of miR-30a was evidenced. In functional assays, miR-30a overexpression attenuated the expression of TGFβ activated kinase1/MAP3K7 binding protein3 (TAB3), α-SMA and fibronectin in both 293T and A549 cells stimulated with TGFβ, in comparison with control culture. The results showed that the downregulation of miR-30a in BALf exosome may be a biomarker for IPF diagnostic, or its overexpression may be an optional treatment for IPF [150]. Meanwhile, Wu et al., in 2020 [151], reported that miR-30a not only regulated the expression of α-SMA, but also regulated the expression of fibroblast activation protein alpha (FAP-α) and COL1A1 in fibroblasts stimulated with TGFβ1. Additionally, their results demonstrated that miR-30a had a role in cell proliferation with or without TGFβ1 treatment, via regulating FAP-α expression. A down-expression of miR-340-5p in various fibrosis related diseases had already been reported. However, its role in IPF is unknown. It had been described previously that the miR-340-5p/activating transcription factor 1 (ATF1) axis regulates the proliferation and invasion of lung-cancer cells [167]. The aim of this study was to investigate if miR-340-5p were associated with the activation of TGFβ-stimulated fibroblasts. The results showed that, once miR-340-5p mimic transfection in normal TGFβ-stimulated fibroblasts, ATF1 was sub-expressed, and the proliferation and activation of these fibroblasts were mitigated. Additionally, they observed a reduction in the COL1A1 and fibronectin protein expression by targeting TGFβ/P38/ATF1 pathway [152].

### MicroRNAs Down-Regulated and Associated with Apoptosis

Although the accumulation of apoptosis-resistant fibroblasts is a hallmark of IPF lungs, the precise mechanisms by which these cells acquire this characteristic have not been fully elucidated yet [104]. For instance, in 2009 it was demonstrated that the miR-29 downregulation in tumoral cells increased the antiapoptotic protein expression and cellular resistance to apoptosis [168]. To date, the contribution of miRNAs in the mechanisms regulating apoptosis susceptibility in IPF is not fully understood. In this context, how the apoptosis intervenes in IPF remains a controversial topic. The miR-29 family has been reported as one of the miRNAs downregulated in fibrotic diseases [49]. The consequence of this downregulation has been linked with an increase in the expression of profibrotic genes associated with the excessive ECM deposition [169,170]. In a study performed in lung fibroblasts of BLM-mice, and in a human fetal lung fibroblast cell line (HFL1), the role of miR-29c as a regulator of lung fibroblasts was evaluated. To attain this objective both miR-29c inhibitor and miR-29c mimic were used. The authors found that the miR-29c mimic inhibits Fas repression induced by TGFβ, restoring the apoptosis sensitivity, a fact that may become an effective strategy for the treatment of IPF [153]. A year later in 2017, Xie et al. [154] also analyzed the miR-29c downregulation function but in lung epithelial cells. The authors observed a decrease in the apoptotic response with a higher viability and proliferation in a mouse lung epithelial cell line (MLE) treated with bleomycin and transfected with the miR-29c mimic, in comparison to MLE control cells. The molecular studies demonstrated that miR-29c suppressed the lung epithelial cells apoptosis by targeting Foxo3, a transcription factor that upregulates cell death and downregulates antiapoptotic proteins. On the other hand, miR-142-3p is a miRNA that has been previously implicated in the cell apoptosis and inflammatory response in fibrosis of cardiomyocytes [171]. However, its role in apoptosis and inflammation in the IPF disease is unknown. In this regard, Guo et al. [155] studied the effects of miR-142-3p mimic or miR-142-3p inhibitor transfection on MLE-12 cells exposed to bleomycin. After the miR-142-3p mimic transfection, an improvement in the cell viability and a reduction in the levels of IL-1 and TNF-α through down-activation of Cox-2 and the P13K/AKT/mTOR signaling pathway were observed. The functional effects with miR-142-3p inhibitor were completely opposite. Previously, it had been reported that miR-506 holds the ability to inhibit EMT targeting Snai2, the E-cadherin transcriptional suppressor [172]. Therefore, the aim of this new study was to evaluate whether miR-506 serves as a regulator of pro-fibrotic factors in the early-stage lung fibrosis; the study used an LPS lung fibrosis mice model. Moreover, the authors specifically focused on target genes of miR-506 involved with the apoptotic and inflammatory responses. The overexpression of this miRNA in the mice model led to the attenuation of LPS-mediated pulmonary fibrosis. The experimental results demonstrated that miR-506 induced the apoptosis directly targeting p65, an essential unit of NF-κB that has a crucial role in the transcriptional responses including cell division, cell survival, differentiation, immunity, and inflammation [173]. It was suggested therefore that miR-506 is a key regulator in pulmonary-fibrosis progression [156]. Another miRNA considered important in fibrosis is MiR-448, which had been pointed out as a tumor suppressor in several types of cancer [174]. However, this miRNA and its target, the ATP-binding cassette subfamily C member 3 (ABCC3), were analyzed to define their role in cell proliferation, apoptosis, and collagen synthesis in the lung fibroblasts of a mice model with IPF. The results obtained showed that upregulation of miR-448 and the inhibition of the expression of its target ABCC3 inhibited fibroblast proliferation and collagen synthesis, but promoted lung fibroblasts apoptosis through of JNK signaling pathway inactivation [157].

## 9. Discussion and Conclusions

Idiopathic pulmonary fibrosis is considered the most lethal of the interstitial lung diseases with an unknown etiology. This illness is characterized by the focal accumulation of activated fibroblast and myofibroblasts derived mainly from EMT, which have the ability to overproduce ECM proteins and alter the epithelial architecture. During these processes, several signaling pathways associated with cell differentiation, migration, and transition are activated. Today, several studies have shown that the dysregulated expression of miRNAs could have an important role in the pathogenesis of IPF. Therefore, updating our knowledge about how these miRNAs intervene in or affect the onset or progression of IPF is an important issue of research. Until now, it has been demonstrated that human miRNAs listed in public databases represent only a small fraction of the total human miRNA repertoire [38,40,175]; therefore, it is necessary to update these studies to improve our understanding about the complex gene expression regulation network in diseases such as IPF. Some studies have estimated that approximately 60% of the human coding transcripts are regulated by miRNAs [38,125,176]. Moreover, recent reports indicate that an important number of miRNAs are associated with the TGF-β-dependent Smad and non-Smad dependent signaling pathway, with the EMT activation process, and with the consequent fibroblast activation and myofibroblast differentiation. Based on the former, evidence suggests that miRNAs may serve as novel therapy for IPF. However, the information obtained from cell lines stimulated with TGFβ or from murine models of pulmonary fibrosis induced with bleomycin is still limited. In addition to the above mentioned, it is unlikely that the classic murine model induced with bleomycin is an adequate source of myofibroblasts during lung fibrosis, this derived from both the rapid onset of symptoms and the partial reversibility of this fibrotic process [177]. Additionally, it is evident that the different origins of the sample under study (lung cell lines, primary culture human fibroblast, or fibrotic murine models) make it difficult to associate the expression levels and role of each different miRNA expressed with the reality of each patient with IPF. Moreover, as if the complexity associated with the pathophysiology associated with IPF was not enough, the TGFβ and associated pathways are not the only involved in the development of this disease. Although most miRNAs reported in this review had already been described in the past, new and noticeable functions have been reported for new target genes, as well as variations in their expression levels. All evidence mentioned supports the sentence: “every miRNA can complement and bind to many different target genes and different miRNAs can also act on the same gene”. However, in this way, and in spite of the complexity where miRNAs are involved, its study and evaluation could open new possibilities for miRNAs to serve as therapeutic agents with more specificity to treat IPF.

Therefore, the absence of actual therapies highlights the need to evaluate other possibilities to treat IPF. Although the use of miRNAs as therapeutic drugs is a possibility yet to be verified in clinical trials, at present, more investigations have focused on the therapeutic role of miRNAs derived from extracellular vesicles (EVs). More studies have investigated the use of miRNAs from mesenchymal stem cells (MSC-EVs) from chronic respiratory diseases; these studies have shown promising results [178,179,180]. It is also important to highlight that at present not only the miRNAs are under evaluation as potential therapeutic targets, but the non-coding RNAs as long non-coding RNAs (lncRNAs) are also being explored for their possible clinical application in respiratory diseases [29,30], which underline the actual relevance of this type of molecule.

## 10. Conclusions

An important topic of research involves updating our knowledge on the role of miRNAs, as well as evaluating their association with the signaling pathways involved with the pathogenesis of Idiopathic Pulmonary Fibrosis. The goal is to improve our understanding of these pathways and molecular mechanisms, still poorly understood, and this improvement could help us suggest both new diagnostic biomarkers and promising strategies for the treatment of IPF disease.

## Figures and Tables

**Figure 1 ijms-23-06613-f001:**
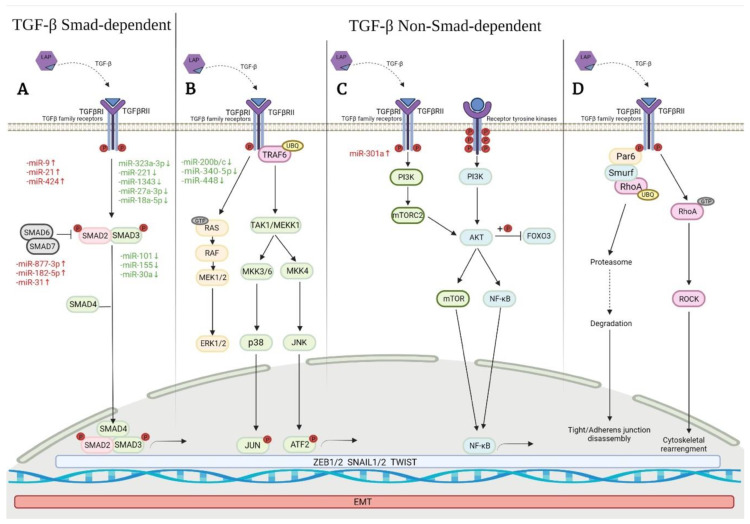
Signaling pathways related to the activation of TGFβ-induced Epithelial Mesenchymal Transition (EMT) in Idiopathic Pulmonary Fibrosis (IPF), and examples of some associated miRNAs. (**A**) Smad signaling TGFβ-induced activates transcription factors that promotes the EMT; (**B**) Non-Smad signaling TGFβ-induced includes the activation of Erk1/2, p38 and JNK MAPK kinases pathways; (**C**) Non-Smad signaling TGFβ-induced activates a receptor with tyrosine kinases activity, which activates both P13K/Akt/mTOR and NF-kβ pathways; (**D**) Non-Smad signaling TGFβ-induced activates ubiquitin-mediated RhoA degradation at tight/adherens junctions or Rho kinase activity during the assembly of actin stress fibers in the TGFβ-induced EMT. Red color indicates up-regulated miRNAs, and green color indicates downregulated miRNAs.

**Figure 2 ijms-23-06613-f002:**
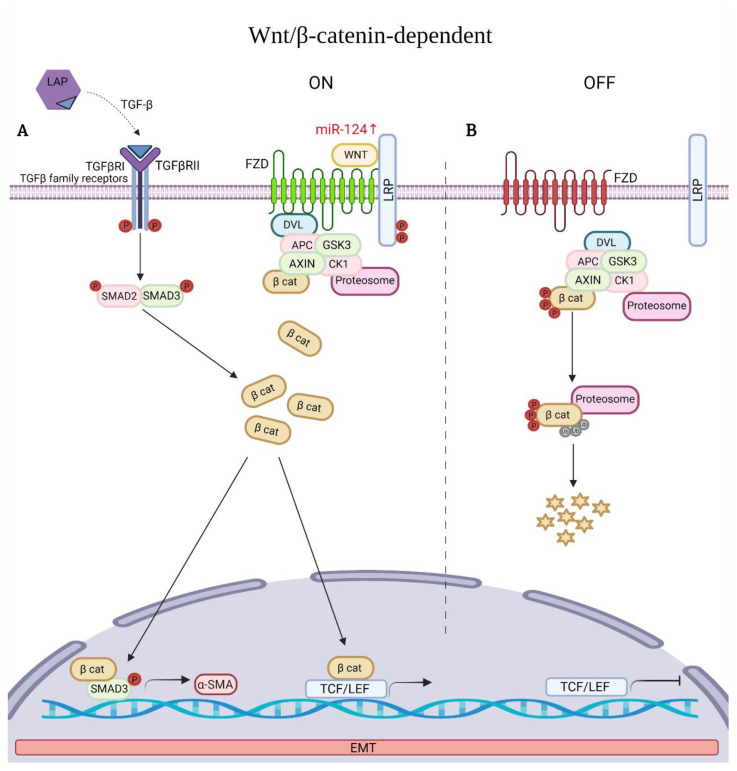
Wnt/β-catenin dependent signaling pathway and an example of associated miRNA (red) in Idiopathic Pulmonary Fibrosis (IPF). (**A**) The binding of Wnt to its receptors Frizzled (FZDs) and to phosphorylated lipoprotein receptor-related protein (LRP) leads to activation of the Wnt signaling pathway. β-catenin is stabilized and translocated in the nucleus, where it binds to T-cell factor/lymphoid enhancer factor-1 (TCF/LEF) to upregulate target genes. TGFβ1 synergizes with nuclear β-catenin regulating alfa-SMA expression. (**B**) In the absence of Wnt, β-catenin is degraded by the proteosome. Red color indicates upregulated miRNA, Stars represent byproducts of β-catenin degradation.

**Table 1 ijms-23-06613-t001:** MiRNAs associated with the pathogenesis of Idiopathic Pulmonary Fibrosis (IPF).

Experimental Model	Mi-RNA	Function	Target of Action	Status in IPF	Effect on IPF and Pathway Associated	Reference
BLM fibrosis mice model and HELF-TGFβind	miR-21	Pro-fibrotic	Smad 2 Smad 3 Smad 4	Up-regulated	TGFβ/Smad pathway activation	[126]
BLM fibrosis rat model	miR-410	Pro-fibrotic	ADAMTS1	Up regulated	↑ECM proteins deposition and fibroblasts proliferation	[127]
LR-MSCs-TGFβind	miR-124	Pro fibrotic	AXIN1	Up regulated	↑Fibrogenic differentiation by Wnt signaling pathway activation	[121]
BLM fibrosis mice model	miR-9	Pro-fibrotic	ANO1	Up regulated	TGFβ-Smad-3 pathway activation and apoptosis suppression	[128]
HFLT, HFLF-TGFβind and LF of BLM fibrosis mice model	miR-301a	Pro-fibrotic	TSC1	Up regulated	Activation of TSC1/mTOR pathway	[129]
NHLF-TGFβind and HFLF	miR-424	Pro-fibrotic	SLIT2	Up regulated	↑myofibroblast differentiation ↑CTGF and α-SMA by TGFβ1/Smad3 pathway	[130]
Mice LR-MSC-TGFβind	miR-152–3p, 140–3p, 148b-3p, 7a-5p _______ miR-34a-5p, 27b-3p, 323-3p, 27a-3p, 34c-5p, 128–3p, 224–5p	Pro-fibrotic _______ Pro-fibrotic	KLF4 ________ ING5	Up regulated _______ Up regulated	Activation of TGFβ and Wnt/β- catenin pathways ____________ Activation of PI3K/AKT and Wnt/β- catenin pathways	[131]
1.2 MiRNAs with pro-fibrotic properties but negative regulators of Smad6/Smad7 expression.						
LR-MSCs-TGFβ and bleomycin mice model	miR-877-3p	Pro-fibrotic	Smad7	Up regulated	↑fibrotic markers and myofibroblasts differentiation induced by TGFβ pathway	[132]
BLM fibrosis mice model and HELF-TGFβind	miR-182-5p	Pro-fibrotic	Smad7	Up regulated	↑pro-fibrotic markers by pSmad2, pSmad3 and TGFβ pathway activation	[133]
A549-TGFβind	miR-31	Pro-fibrotic	Smad6	Up-regulated	TGFβ Smad2 activation	[134]
1.3 MiRNAs with anti-fibrotic properties and with regulatory activity of signaling pathways linked with IPF.						
A549-TGFβind	miR-184	Anti- fibrotic	Smad2/Akt	Down-regulated	TGFβ-P13k-AKt pathway activation	[134]
HFLT and BLM fibrosis mice model	miR-323a-3p	Anti-fibrotic	TGFα Smad2	Down regulated	Activation of both TGFα and TGFβ signaling pathways	[135]
BLM fibrosis mice model	miR-29b	Anti-fibrotic	COL1A1 and COL3A1	Down-regulated	↑collagen expression	[136]
Human BAL cells and THP-1 cells	miR-185 miR-29a	Anti-fibrotic Anti- fibrotic	AKT COL1A1	Down regulated Down Regulated	↑collagen expression and ↑of EMC proteins deposition	[137]
Adenocarcinoma epithelial cell lines and HFLT	miR-185 and miR-186	Anti-fibrotic	COL5A1	Down-regulated	↑EMT process	[138]
HFLT	miR-130b-3p	Anti-fibrotic	IGF-1	Down regulated	↑Collagen1A1 expression and ↑proliferation and migration of fibroblasts	[139]
HFLT, A549, HBEC and BLM fibrosis mice model	miR-221	Anti-fibrotic	HMGA2	Down regulated	↑EMT and fibrotic markers by activation of TGFβ1/Smad3 pathway	[140]
A549-TGFβind NHLF-TGFβind	miR-1343	Anti-fibrotic	TGFβRI and TGFβRII	Down regulated	Activation of TGFβ pathway and ↑expression of fibrotic markers and EMT	[141]
HFLF and NHLF	miR-27a-3p	Anti-fibrotic	Smad2 Smad4 and α-SMA	Down Regulated	↑ myofibroblasts differentiation by TGFβ pathway activation	[142]
Lung and LF of BLM fibrosis mice model	miR-27b	Anti-fibrotic	TGFβR1 and Smad2	Down Regulated	↑fibrotic markers by TGFβ pathway activation	[143]
PMCs	miR-18a-5p	Anti-fibrotic	TGFβRII	Down regulated	↑EMT through TGFβ1 Smad2/3 complex	[144]
LL29, NHLF and BLM mice model	MiR-101	Anti-fibrotic	Col1A1 Col3A1 α-SMA	Down regulated	↑fibroblast proliferation and activation via TGFBR1	[145]
NHLF-TGFβind	MiR-155	Anti-fibrotic	Smad1	Down regulated	↑fibroblast proliferation, migration, and collagen synthesis by TGFβ activity	[146]
LPS early pulmonary fibrosis mouse model	miR-200b/c	Anti-fibrotic	ZEB1/2	Down regulated	Activation of EMT via p38 MAPK and TGF-β/Smad3 pathway	[147]
LF of BLM and silica fibrotic mice model	miR-19a- 19b-20a sub-cluster	Anti-fibrotic	TGFβRII	Down regulated	Activation of TGFβ pathway ↑expression of pro-fibrotic genes	[148]
NHLF-TGFβind	miR-133a	Anti-fibrotic	COL1A1 CTGF α-SMA TGFβR1	Up-regulated	Functions as a feed-back negative regulator of TGFβ pathway	[149]
293T-TGFβind A549-TGFβind	miR-30a	Anti-fibrotic	TAB3 αSMA FNT	Down regulated	Activation of TGFB pathway	[150]
MRC-5 cells	miR-30a	Anti-fibrotic	FAP-α	Down regulated	↑FAP-α, col1a and α-α-SMA synthesis	[151]
NHLF-TGFβind	miR-340-5p	Anti-fibrotic	FNTATF1	Down- regulated	↑TGFβ/P38/ATF1 pathway	[152]
1.4 MicroRNAs down-regulated and associated with Apoptosis.						
LF cell lines and LT of BLM-fibrotic mice model	miR-29c	Anti-fibrotic	PARP-1 LOXL2 COL3A1 SPARC	Down-regulated	↑Resistance to Fas-mediated apoptosis and ↓activity of caspase-8/caspase-3/PARP-1 pathway	[153]
AEC2s of IPF, healthy LT and MLECs	miR-29c	Anti-fibrotic	Foxo3a	Down regulated	↑apoptosis associated with reduced epithelial cell renewal	[154]
MLE-12 cells plusBleomycin	miR-142-3p	Anti-fibrotic	Cox-2	Down regulated	↑apoptosisP13K/AKT/mTOR pathway inactivation	[155]
LPS-lung fibrosis mice model	MiR-506	Anti-fibrotic	p65 (NFκβ subunit)	Down regulated	Apoptosis resistance	[156]
LF of BLM fibrosis mice model	miR-448	Anti-fibrotic	ABCC3	Down regulated	Apoptosis resistance by JNK signaling pathway ↑proliferation and collagen synthesis	[157]

Abbreviations: ADAMTS = A Desintegrin and Metalloproteinase with Thrombospondin motifs; AEC2 = Alveolar epithelial cell type II; ATF-2 = Activating Transcription Factor 2; BAL = Bronchioalveolar Lavage; BLM = Bleomycin; CTGF = Connective Tissue Growth Factor; ECM = Extracellular Matrix; EMT = Epithelial Mesenchymal Transition; FAP = Fibroblast Activation Protein; FNT = Fibronectin; HBEC = Human Bronchial Epithelial Cell; HELF = Human Embryonic Lung Fibroblasts HFLF = Human Fibrotic Lung Fibroblasts; HFLT = Human Fibrotic Lung Tissues; IGF = Insulin Growth Factor; IPF = Idiopathic Pulmonary Fibrosis; LF = Lung Fibroblasts; LT = Lung tissue; LPS = Lipopolysaccharide; LR-MSCs = Lung Resident Mesenchymal Stem Cells; TGFβind = Induction by Transforming Growth Factor beta; MLECs = Mouse lung epithelial cells; MMP = Matrix Metalloproteinases; MLE = Mouse Lung Epithelial cells; NHLF = Normal Human Lug Fibroblasts; PMCs = Pleural Mesothelial Cells.

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
