# Peer review of "Role of MicroRNAs in Signaling Pathways Associated with the Pathogenesis of Idiopathic Pulmonary Fibrosis: A Focus on Epithelial-Mesenchymal Transition"

_ijms, 2022, doi:10.3390/ijms23126613_

Round 1
Reviewer 1 Report
This is an interesting review presenting the role of an emerging set of small molecules, namely non-coding RNA in the regulation of the IPF modulatory machinery
With the intent to give completeness to the nice work done by the authors I have some suggestions that should be incorporated into the manuscript
1. In the introduction, references concerning the potential IPF etiopathogenesis are poor and important mechanisms potentially involved in IPF onset and progression are missing. Please, see this recent comprehensive review concerning the cellular and molecular determinants of IPF (PMID: 33201251)
Authors mainly focused on works concerning EMT, fibroblast differentiation, and synthesis of ECM, however other important mechanisms should be discussed such as EndMT, Reactive Oxygen Species, and Oxidative Stress for instance. As an example, see the following paper (PMID: 35189048). I am quite sure there are many papers concerning ROS, Lung Fibrosis, and non-coding RNA
Reviewer 2 Report
In this manuscript the Authors report the most recent knowledge about the role of micro-RNA in the pathogenesis of IPF. The review seems comprehensive and is clear enough even for the reader without experienced in this topic.
Comments:
- The title should be changed to make it clear that it is a revision (narrative or systemic)
- In the abstract it should be explained how the publications were selected
- L641 should be considered -> is
Round 2
Reviewer 1 Report
This reviewer would like to thank the authors for considering the suggested points. Indeed, the authors comprehensively amended the introduction, adding the majority of the known mechanisms involved in the pathogenesis of lung fibrosis. Nonetheless, this review is mainly focused on the epithelial-mesenchymal transition (EMT) and miRNA interaction, and for this reason, I believe the title should be changed to highlight this aspect.
I suggest "Role of microRNAs in signaling pathways associated with the pathogenesis of Idiopathic Pulmonary Fibrosis. A focus on epithelial-mesenchymal transition"
Author Response
"Please see the attachment"
